# Removal of Carbamazepine in Aqueous Solution by TiO₂ Ceramic Photo-Catalyst under Simulated Solar Light: Kinetics, Effects of Environmental Factors and Degradation Pathways

**Trinh Hoang Nghia** [1], **Vu Toan Khanh** [1], **Cam Tu Vu** [1], **Nguyen Thi Kieu Oanh** [1], **Nguyen Thi Van Anh** [1], **Le Hong Luyen** [1], **Nobuaki Negishi** [2], **Sylvain Ouillon** [1,3] **and Bui Van Hoi** [1,*]

[1]  Department of Water–Environment–Oceanography (WEO), University of Science and Technology of Hanoi (USTH), Vietnam Academy of Science and Technology (VAST), 18 Hoang Quoc Viet, Cau Giay, Hanoi 122100, Vietnam; kingoffeeders@gmail.com (T.H.N.); vutoankhanh99@gmail.com (V.T.K.); vu-cam.tu@usth.edu.vn (C.T.V.); nguyen-thi-kieu.oanh@usth.edu.vn (N.T.K.O.); nguyen-thi-van.anh@usth.edu.vn (N.T.V.A.); le-hong.luyen@usth.edu.vn (L.H.L.); sylvain.ouillon@legos.obs-mip.fr (S.O.)

[2]  Environmental Management Research Institute, National Institute of Advanced Industrial Science and Technology, 16-1 Onogawa, Tsukuba 305-8569, Japan; n-negishi@aist.go.jp

[3]  UMR LEGOS, University of Toulouse, IRD, CNES, CNRS, UPS, 14 Avenue Edouard Belin, 31400 Toulouse, France

\*  Correspondence: bui-van.hoi@usth.edu.vn; Tel.: +84-982-975-883

**Abstract:** Current research on the photocatalytic activity of TiO₂ mainly focuses on its nano- or micro-particle forms, which are difficult to recycle and apply in real engineering applications. This study aims to apply a small pilot of TiO₂ in the ceramic form to remove carbamazepine (CBZ) from an aqueous solution under simulated sunlight. A high removal efficiency up to >99% was shown in a 5 mg L⁻¹ CBZ solution after 6 h of irradiation with a total energy of 150.92 kJ. The kinetic degradation was not affected in an alkaline solution (at pH 7, pH 10, and pH 13) but was faster under acidic conditions (pH 2) in which CBZ existed in the protonated form. The presence of NO₃⁻ (10–50 mg L⁻¹) slightly affected the photodegradation of CBZ while humic acid significantly reduced the photocatalytic activity. In addition, the presence of major ions in water also had a negative effect at concentrations between 10 and 50 mg L⁻¹. The MS/MS was used to identify the transformation products of CBZ, and a possible degradation mechanism was proposed. The toxicity of CBZ and the by-products was primarily evaluated. The results showed that TiO₂ ceramics show high reusability and stability with a photocatalytic performance of >95% and a mass loss of <5% after 90 degradation cycles.

**Keywords:** carbamazepine; water treatment; photocatalysis; photo-degradation; TiO₂ ceramic; MS/MS identification

## 1. Introduction

Over the last two decades, pharmaceutical residues, as new emerging pollutants, have received increasing attention due to their high persistence in wastewater treatment and their ubiquitous detection frequency in various water systems [1–3]. One of the famous pharmaceuticals widely used in treating psychomotor attacks is carbamazepine ($C_{15}H_{12}N_2O$, hereafter noted as CBZ) with a consumption of up to 1014 tons per year [4]. CBZ was one of the most pharmaceutical residues found in an aquatic environment. Its concentration was reported to be up to 21.6 μg L⁻¹ in the influents of a wastewater treatment plant [5] and less than 1 μg L⁻¹ in natural surface water [6]. In the environment, CBZ can affect embryonic cells, preventing their development and causing defects. In some cases, low doses of CBZ have a strong toxic effect on the central nervous system and the digestive system. Long-term use of CBZ can lead to changes in blood cells and lipids, a decrease in white blood cell levels, and even severe symptoms [7–9]. In addition, carbamazepine is also

an aromatic nitrogen heterocyclic component with a symmetrical structure, which cannot be efficiently eliminated by biochemical treatment processes. Recently, different types of materials have been developed to remove CBZ, such as $BiOCl/Fe_3O_4$ [10], $g\text{-}C_3N_4/TiO_2$ composites [11], $Fe_3O_4/BiOBr/BC$ [12], or self-assembled $EHPDI/TiO_2$ [13]. However, these materials are not easy to synthesize and are difficult to implement in real applications because they are usually present as fine particles that require a great amount of energy to recover the catalyst after use. Titanium dioxide ($TiO_2$) is the most popular material used in advanced oxidation processes (AOPs) due to its high ability to degrade chemical pollutants and pathogens, including bacteria and fungi. In addition, the band gap of $TiO_2$ is between ~3.0 eV (rutile) and ~3.2 eV (anatase), within which it can be activated by sunlight. The combination of sunlight and photocatalysis can significantly reduce processing costs and is environmentally friendly. In this field, most of the recent studies have been conducted to investigate the photocatalytic behavior of $TiO_2$ in various forms of nanoparticles [14], slurry power [15], or composites [16]. Recently, $TiO_2$-based coatings [17] and hybrid nanomaterials [4,18] have shown promise as high performance photocatalysts for removing emerging contaminants in an aqueous solution. Micro- and nanostructured $TiO_2$-based or hybrid materials were shown to have a high removal efficiency for CBZ in a shorter time, but an additional filtration separation step was added to the conventional steps (washing, filtering, and drying) [19–21]. Therefore, these new materials were still used only at the laboratory scale. Furthermore, a decrease in photocatalytic activity was observed in previous works for only five cycles [11,18,19]. This proves their instability and shows the need to improve them before large scale application. In addition, although these materials showed a high CBZ removal efficiency of up to 100%, mineralization was only 60–70%. Therefore, by-product identification is necessary to study the degradation mechanism as well as to evaluate the toxicity of the by-products using a Vibrio fischeri test or evaluating them mainly with available toxicity software.

In this context, the present study investigated the photocatalytic performance of $TiO_2$ in the form of high-durability ceramics underwater. The main advantages of this $TiO_2$ ceramic are that it is efficient, cheap, and easy to use and maintain. This ceramic was successfully synthesized and characterized by Kato et al., 2020 [22] and used for CBZ degradation under sunlight with various conditions regarding pH, water matrices, humic acid, and major ions in water. The photodegradation efficiency of CBZ was determined by an HPLC-DAD system, while the by-products were identified using an MS/MS system. The degradation mechanisms are also proposed. Finally, the software ProTox-II was used to evaluate the toxicity of the by-products.

## 2. Materials and Methods

### 2.1. Chemical Reagents

Titanium dioxide ($TiO_2$), the photocatalytic material, was used in the form of ceramic (Solid flake PSF-001), obtained from photocatalytic materials incorporated by Sowaku, Nagoya, Japan. The synthesis and characterization of this material were described by Kato et al., 2020 [22]. At first, the precursor monolith was prepared by mixing a titanium colloidal sol (PT-01, 15–20% of Ti solid) with activated carbon (ratio: 200/1 *w/w*). The mixing gel was then poured into a mold and dried to obtain the monolith titania gel, which was then calcined at 525 °C to a final $TiO_2$ ceramic with length from 2 to 8 mm and thickness of 1 mm. The composition of $TiO_2$ included anatase, brookite, and rutile, which was confirmed by XRD and Raman spectra. The specific surface area was approximately 100 $m^2/g$, which was higher than that of P25 (standard photocatalyst powder). Carbamazepine ($C_{15}H_{12}N_2O$, purity 99.78%), humic acid (HA), $KNO_3$, $NaHCO_3$, $NH_4Cl$, $CaCl_2$, NaOH, and concentrated HCl (37%) were purchased from Sigma-Aldrich, Singapore. Solvents (methanol and acetonitrile) of analytical HPLC grade were purchased from Fisher Scientific (Singapore). The ultrapure water was produced from Smart2pure 12 UV (Thermo Scientific, Waltham, MA, USA).

## 2.2. Solution Preparation

The CBZ solution (5 mg L$^{-1}$) was prepared daily by dissolving 5 mg CBZ in one liter of variable water (ultrapure, tap, mineral, sea, and river water). HA, cations, and anions were added to the solution in the 5–50 mg L$^{-1}$, except for Ca$^{2+}$ and Mg$^{2+}$, which were chosen as hard water concentrations according to Vietnam Standards (60–180 mg L$^{-1}$). In addition, the pH was adjusted by adding HCl 0.1 M or NaOH 0.1 M using a pH meter (Sension + pH3 Basic laboratory pH and ORP Meter from Hach, Loveland, CO, USA).

Surface water samples used in this experiment were collected from the Red River and an urban lake in Hanoi city. After sampling, the water samples were filtered through a GF/F membrane (glass microfiber filter, 0.7 μm). The TOC values were measured as 2.099 mg L$^{-1}$ and 3.462 mg L$^{-1}$. Mineral composition of Na$^+$, Mg$^{2+}$, and Ca$^{2+}$ was 57.186 mg L$^{-1}$, 12.518 mg L$^{-1}$, and 16.403 mg L$^{-1}$, respectively, in the river sample, and 16.386 mg L$^{-1}$, 4.98 mg L$^{-1}$, and 11.381 mg L$^{-1}$, respectively, in the lake sample.

Sea water was collected from the Gulf of Tonkin with the major compositions of Na$^+$: 5386 mg L$^{-1}$, Mg$^{2+}$: 99.3 mg L$^{-1}$, Ca$^{2+}$: 60.2 mg L$^{-1}$, and Fe$^{2+}$: 0.8 mg L$^{-1}$, which were determined by ICP-MS (iCap, Thermo Scientific). This water was also filtered through a GF/F membrane to remove suspended particles. The salinity of this sample was 13.7 and the conductivity was 22.2 mS cm$^{-1}$. Its TOC value was 0.881 mg L$^{-1}$.

Tap water was directly collected at the laboratory with the following dominant mineral composition: Na$^+$: 2.395 mg L$^{-1}$, Mg$^{2+}$: 3.9 mg L$^{-1}$, and, Ca$^{2+}$: 17.015 mg L$^{-1}$. The TOC value of the tap water used was 0.399 mg L$^{-1}$.

## 2.3. Experimental Setup

A 40 g photocatalyst (TiO$_2$) was packed in 4 × 20 cm Pyrex glass tubes (inner diameter: 1 cm). The reaction tubes were installed in the simulated sunlight reactor (model SUNTEST CPS+, Atlas Material Testing Technology LLC, Mt Prospect, IL, USA) equipped with a 400 W/m$^2$ irradiated xenon lamp in the 300–800 nm range (Figure 1). The total energy dosage that was recorded by the SUNTEST CPS+ system was 21.56 kJ for 6 h of irradiation. The simulated light system was also installed with a daylight filter to limit the radiation under 290 nm (UV special glass, code: 56052371 UV). One liter of 5 mg L$^{-1}$ CBZ (variable conditions) with various environmental factors such as pH, humic acid (HA), anions, and cations was circulated through the degradation system at a rate of 50 mL min$^{-1}$ for 6 h. The CBZ solution was circulated through a photocatalyst (TiO$_2$) for 30 min in the dark to reach an adsorption/desorption equilibrium. Then, the solution was irradiated, and samples were taken out every 30 min to analyze the removal efficiency of CBZ. All experiments were conducted at room temperature and the reactor was set at 25 °C.

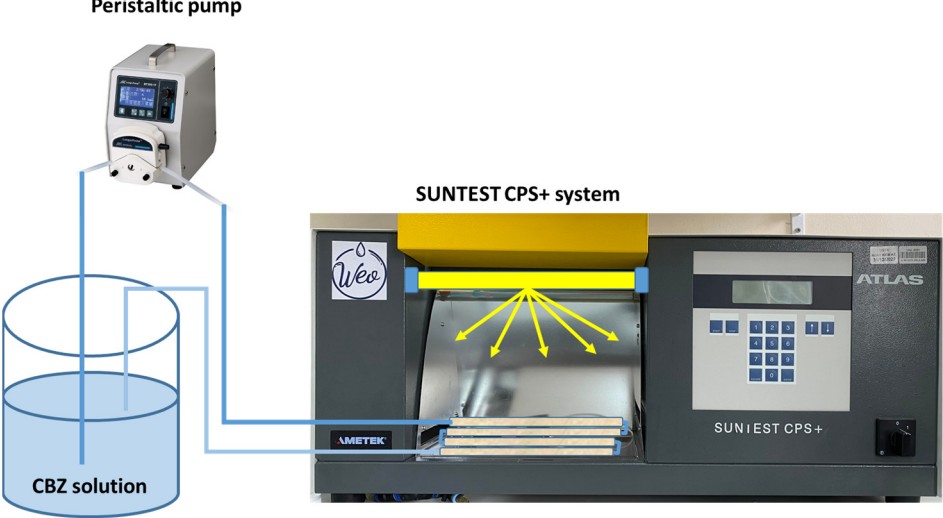

**Figure 1.** The degradation system of carbamazepine.

*2.4. Analysis*

2.4.1. High-Performance Liquid Chromatography (HPLC)

The concentration of CBZ was measured by an HPLC-DAD (Vanquish Core, Thermo Scientific). The separation of CBZ and its by-products (BDS, C18, 100 × 2.1 m products) was performed on a reserved phase C18 column where the column chamber was constantly set at 40 °C. The detection wavelength was 240 nm and the injection volume was 20 μL. The mobile phase consisted of ultrapure water (phase A) and MeOH (phase B). The gradient started at 90% of phase A and decreased linearly to 10% at 8 min. This composition was maintained for 3 min before returning to the initial condition (Table 1). This state was maintained isocratically until 15 min to make the column pressure constant and await the next injection. The flow rate was kept constant at 0.4 mL min$^{-1}$. The retention time of carbamazepine was 7.7 ± 0.1 min.

**Table 1.** Gradient for the mobile phase of HPLC.

| Time (min) | Flow Rate (mL/min) | % UPW | % MeOH |
|:---:|:---:|:---:|:---:|
| 0 | 0.4 | 90 | 10 |
| 8 | 0.4 | 10 | 90 |
| 11 | 0.4 | 10 | 90 |
| 11.1 | 0.4 | 90 | 10 |
| 15 | 0.4 | 90 | 10 |

2.4.2. Total Organic Carbon Analyzer (TOC)

In this research, Shimadzu TOC-V CSN (Shimadzu Corporation, Kyoto, Japan) was used to determine the mineralization during the experiment. The combustion tube with platinum catalyst was set at 680 °C and an oxygen flow (5.0 grade) was used as the flue gas. The calibration curve was prepared daily from potassium hydrogen phthalate ($C_8H_5KO_4$) in the 0.4 to 6.67 mg L$^{-1}$ carbon range. The sensitivity of the instrument was checked daily using the solution (2 mg L$^{-1}$). All samples were acidified with 50 μL of concentrated HCl (37%) and the acidified samples were then purged with ultrapure oxygen for 90 s to remove inorganic carbon before injection. Each measurement was injected with a 100 μL sample, which was repeated 3 times, and each injection lasted 5 min.

2.4.3. Identification of By-Products

By-products were identified by direct injection into an MS/MS system (XevoTQD, Waters, Milford, MA, USA) with the full scan mode ($m/z$: 50 to 450). The samples were directly injected into the system with a flow rate of 100 μL min$^{-1}$. The ionization was performed with an electrospray ionization (ESI) source in positive mode. The source parameters were set as follows: desolvation gas, 1000 L h$^{-1}$; cone gas, 10 L h$^{-1}$; desolvation gas temperature, 500 °C; source temperature, 150 °C; capillary voltage, 3000 V. The complete spectrum mass ($m/z$) was compared to previous works to identify the by-products and propose the degradation mechanism.

*2.5. Toxicity of By-Products*

In this experiment, ProTox-II software was used to predict the toxicity of CBZ, and all by-products were created after the degradation process [3,23]. The ProTox-II combined different parameters, such as molecular similarity and pharmacophore, and used machine learning to predict the various toxicities (hepatotoxicity, cytotoxicity, carcinogenicity, immunotoxicity, etc.). ProTox-II was freely accessed at https://tox-new.charite.de/protox_II/ (last updated on 1 February 2021).

## 3. Results and Discussion

### 3.1. Photocatalytic Test and Kinetic Degradation

### 3.1.1. Effect of pH

The surface charge of the photocatalyst and the speciation of CBZ depend on the pH value. CBZ has two $pK_a$ values (2.3 and 13.9) [24]. Therefore, CBZ is favored to exist in the neutral form in pH ranging from 3 to 11, while at pH 13, the main form of CBZ is anionic. These experiments were performed in ultrapure water (pH 7) and the pH was adjusted by adding a small amount of HCl 0.1 M or NaOH 0.1 M monitored by a pH meter. The degradation efficiency of CBZ by simulated sunlight/TiO$_2$ photocatalyst under different pH values is shown in Figure 2. The degradation reaction rates at different conditions were shown in Table S1. The reaction rate constants were determined by plotting semi-log graphs of the initial concentration/interval concentration at time t versus the interval time. In all cases, the correlation coefficient values ($R^2$) were higher than 0.95, which indicated that all reaction rates followed pseudo-first-order kinetics. CBZ degradation and the reaction rate constant were highest at pH 2 and CBZ was completely removed after 180 min. As the pH increases, the percentage removal significantly decreased. The degradation efficiencies at pH 7 and 10 were 84.9% and 83.6%, respectively, after 180 min of irradiation and the value of k (the reaction rate constant) decreased from 0.0253 min$^{-1}$ at pH 2 to 0.0101 min$^{-1}$ at pH 10. These results were similar to those reported by Chen et al., [25] for the photodegradation of CBZ by photocatalyst BiOCl/Fe$_3$O$_4$. It was proposed that the contribution of photo-generated h$^+$ (Equation (1)) to CBZ degradation was about 95.0%, while the effect of HO$^\bullet$ radicals on the process was only 3.2% [25] because h$^+$ could be more oxidizing than HO$^\bullet$ (the oxidation potential of TiO$_2$ is about 2.9 eV higher than that of HO$^\bullet$, which is 2.38 eV). Thus, when the pH increased to 10, the concentration of OH$^-$ also increased, which led to the decrease in the concentration of h$^+$ according to the reaction (Equation (2)), so the efficiency of CBZ degradation decreases when the amount of h$^+$ decreases (at higher pH).

At 3 < pH < 10, CBZ is in the neutral form [8]. The surface of the photocatalyst (TiO$_2$) becomes positively charged in a solution with lower pH because the pZc of the TiO$_2$ photocatalyst is about 5–6 [26]. This increased the interaction between the electron-rich aromatic nucleus of CBZ and the photocatalyst, and the degradation efficiency was most effective in lower pH solutions [27]. This was even more obvious when the pH increased from 10 to 13, which changed the state of CBZ from its natural to anionic form, thus increasing the decomposition capacity. However, with a sudden increase in pH, the concentration of h$^+$ decreased relatively, resulting in the degradation capacity obtained after 6 h of experimentation only reaching nearly 60% compared to 100% at pH 2 or pH 7. In addition, the value of k at pH 13 is only 0.0023, which is five times lower than at pH 10 or pH 7.

$$TiO_2 \rightarrow TiO_2 \ (e^- + h^+) \tag{1}$$

$$TiO_2 \ (h^+) + OH^- \rightarrow TiO_2 + HO^\bullet \tag{2}$$

### 3.1.2. Effect of Anions and Humic Acid (HA)

The effects of major anions, such as chloride (Cl$^-$), nitrate (NO$_3^-$), bicarbonate (HCO$_3^-$), carbonate (CO$_3^{2-}$), and humic acid (HA), existing in the environment were studied. The decomposition rate increased slightly in the presence of 50 mg L$^{-1}$ NO$_3^-$, while it significantly decreased with the presence of individual anions, including HCO$_3^-$, Cl$^-$, CO$_3^{2-}$, and humic acid (HA), at 50 mg L$^{-1}$ (Figure 3). A depletion in the concentration of photo-generated h$^+$ via its combination with other anions, including Cl$^-$, HCO$_3^-$, HA, and CO$_3^{2-}$ (Equations (3)–(6)), prevents h$^+$ from degrading CBZ directly [25]. Thus, with the addition of HA, Cl$^-$, and HCO$_3^-$ at low concentrations of 10 mg L$^{-1}$ and 20 mg L$^{-1}$, the degradation efficiency in UPW decreased slightly from 100% to 92.7%, 88.5%, and 95.8%, respectively, at 10 mg L$^{-1}$ and from 100% to 97.1%, 94.0%, and 92.1%, respectively, at 20 mg L$^{-1}$. With an increase in the concentration of HA, Cl$^-$, or HCO$_3^-$ to 50 mg L$^{-1}$,

these values were only 81.05%, 91.17%, and 84.06%, respectively. This shows the negative effect of humic acid and these two inorganic anions on the decomposition process of CBZ.

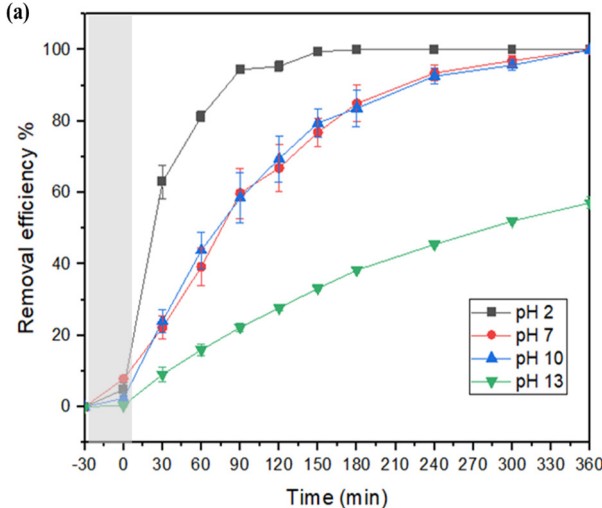

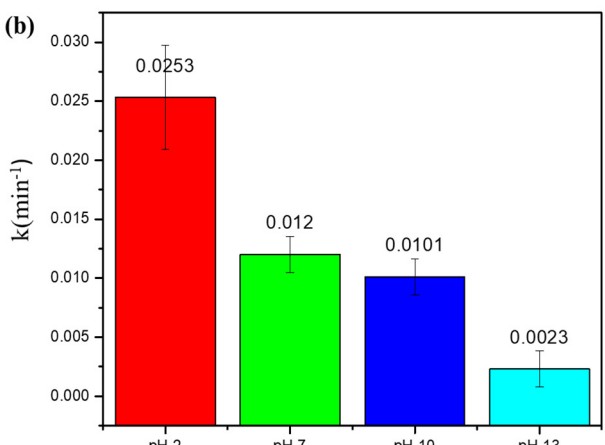

**Figure 2.** Degradation of CBZ in solutions of different pH: (**a**) Removal efficiency; (**b**) kinetic degradation.

Although the concentration of selected $CO_3^{2-}$ was different from that of the previous anions, the main tendency was still to hinder the decomposition process of CBZ because $h^+$ is acted by external anions, resulting in a low decomposition efficiency. However, as the concentration of the selected anion was also several times higher, the efficiency of the decomposition process was also significantly reduced, reaching only 71.5%, 70.1%, and 56.7% for $CO_3^{2-}$ concentrations of 50, 100, and 200 mg $L^{-1}$, respectively. In contrast, the addition of $NO_3^-$ slightly enhanced the CBZ photodegradation, which could be explained by the formation of two $NO_2^{\bullet}$ and $HO^{\bullet}$ radicals under simulated solar irradiation (Equation (7)) [6]. The efficiency of photodegradation with the addition of $NO_3^-$ ions (from small to large amount) was slightly higher than under UPW conditions throughout the experiment. For example, after 180 min, the photodegradation efficiency with $NO_3^-$ additions of 10, 20, and 50 mg $L^{-1}$ was 81.8%, 86.1%, and 85.5%, respectively, which is slightly higher than that of the samples without $NO_3^-$ (80.9%). In addition, the degradation efficiency also reached 100% after 6 h, showing that CBZ was also completely decomposed. These results confirm the enhancing effect of $NO_3^-$ photodegradation.

The presence of humic acid was perhaps the most questionable because many previous studies have shown that the addition of HA to the original solution could accelerate the decomposition process since HA could produce other reactive species such as singlet oxygen

($O_2$), superoxide anions ($O_2^{\bullet-}$), and solvated electrons that enhanced the efficiency of degradation [21,28,29]. For example, the photodegradation efficiency of 17$\alpha$-ethynylestradiol (EE2) was four times faster than in ultrapure water in the presence of 5 mg L$^{-1}$ HA [30]. However, it was observed that the photodegradation of CBZ decreased in the presence of HA. The degradation efficiency of CBZ in the presence of 50 mg L$^{-1}$ HA was only 55.6% and 81.1% after 180 min and 360 min of irradiation, respectively. The rate constant decreased from 0.0109 to 0.0044 min$^{-1}$. These results are similar to previous studies [10,12]. Similar to the effects of the anion when present in the original carbamazepine solution, HA also captured the photo-generated h$^+$ to form the (HA$^{\bullet+}$) radical [11], leading to the decrease in h$^+$ in the solution, and the degradation efficiency also dropped significantly. Moreover, humic substances could generate reactive species such as solvated electrons, hydroxyl radicals, singlet oxygen, and reactive triplet states with photochemical excitation. That was the main reason for gradually reducing the decomposition of carbamazepine [12].

$$HCO_3^- + h^+ \rightarrow HCO_3^{\bullet} \tag{3}$$

$$CO_3^{2-} + h^+ \rightarrow CO_3^{\bullet-} \tag{4}$$

$$Cl^- + h^+ \rightarrow Cl^{\bullet} \tag{5}$$

$$HA + h^+ \rightarrow HA^{+\bullet} \tag{6}$$

$$NO_3^- + H_2O \rightarrow NO_2^{\bullet} + OH^- + HO^{\bullet} \tag{7}$$

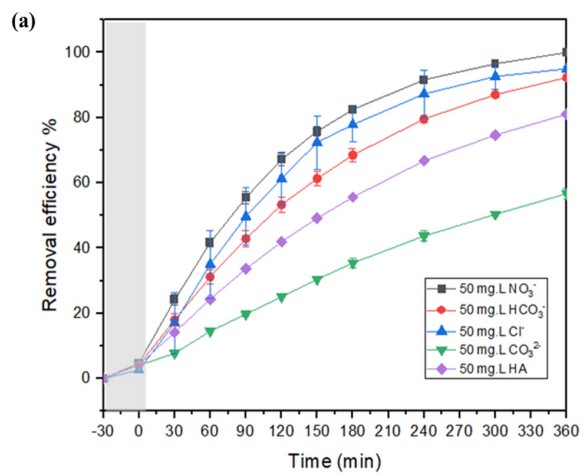

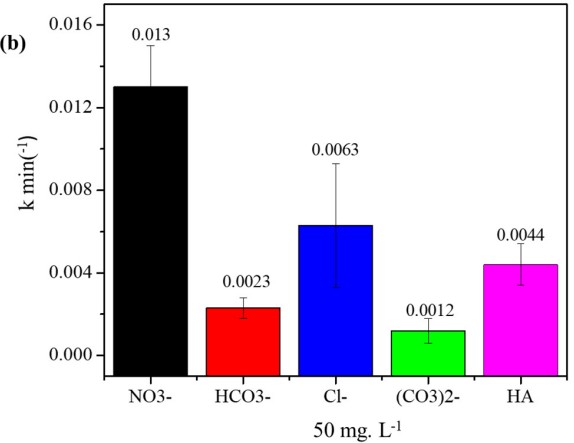

**Figure 3.** Degradation of CBZ with the addition of anions and HA: (**a**) Removal efficiency; (**b**) kinetic degradation.

### 3.1.3. Effect of Cations $Ca^{2+}$, $Mg^{2+}$, and Water Matrices

The effect of $Ca^{2+}$ and $Mg^{2+}$, which are major cations in surface water, was also investigated. With the presence of $Ca^{2+}$ and $Mg^{2+}$ in the solution, precipitation was observed during the experiment (Figure 4). With higher cation concentrations, more precipitate was observed on the surface of the catalyst and the reaction tube wall (Equation (8)).

With the accumulation of white precipitates on the photocatalyst surface, it was expected that the photocatalytic performance of $TiO_2$ would be reduced with time. However, the results showed the opposite trend (Figure 4). After 9 cycles and gradually increasing the concentration of both cations (from 60 mg $L^{-1}$ to 180 mg $L^{-1}$), the degradation efficiency of CBZ after 6 h of testing did not change significantly compared to the experiment without the two cations (from 100% to 96%). This was also reported by Negishi and colleagues who concluded that, despite the accumulation of precipitation on the catalyst, the photocatalytic activity only slightly decreased [31].

Figure 4 also shows the photodegradation efficiency of CBZ in different water matrices: lake, river, sea, tap, and ultrapure water. Similarly to the previous results, $TiO_2$ can remove CBZ in these matrices under solar irradiation. In general, CBZ is degraded most effectively in UPW, while $TiO_2$ has the worst catalytic performance in lake or river water. This could be explained by the fact that although decomposition can still occur efficiently, complete degradation can be suppressed in the presence of some dissolved organic matter such as HA in lake or river water [32].

The CBZ decomposition process that occurred in the seawater and tap water matrix was also studied to evaluate the impact of these two types of water environments on the ability of $TiO_2$ materials to remove CBZ. The lower degradation efficiencies in seawater and tap water matrices illustrated the detrimental effect of natural waters with the presence of higher concentrations of inorganic salts, dissolved organic matter, etc. $HO^\bullet$ radicals or photo-generated $h^+$ (strong oxidizing agents) are thus transferred to weaker oxidizing agents such as $Cl^{\bullet-}$ and $CO_3^{\bullet-}$ [33], leading to a decrease in the degradation efficiency of CBZ (Equations (9)–(11)).

The degradation rate of CBZ in seawater was higher than in fresh surface water, even though the concentration of $Mg^{2+}$ and $Ca^{2+}$ is higher in seawater than in surface water. This phenomenon could be explained by the fact that the presence of $Cl^-$ at high concentrations can prevent the adsorption of reaction inhibitors ($HCO_3^-$ and $CO_3^{2-}$) on $TiO_2$. The results showed that the decomposition ability of CBZ reached high to low values in ultrapure water, tap water, seawater, river water, and lake water. The degradation efficiency of CBZ after 6 h in ultrapure water reached 100%, whereas this value in other waters tended to drop by 10–20% (Figure 4). These experiments showed that HA led to a significant decrease in CBZ degradation because it not only reduced strong oxidizing agents, such as other anions ($Cl^-$ and $HCO_3^-$), but also competed directly with CBZ degradation.

The TOC value of water samples was determined before CBZ decomposition. In river and lake waters, the TOC was 8–10 times higher than in seawater samples, showing that the HA concentration in the river or lake samples were higher, resulting in a decrease in CBZ degradation (from 96.2% to 86.5% and 80.8% in the river and lake samples, respectively) [34]. Additionally, the TOC value was more than twice as high when comparing seawater samples to tap water samples, and the concentration of cations that interfered with the decomposition process in seawater samples was five times higher than that of tap water, so that, after 180 min, the decomposition capacity of CBZ in seawater was only 70.9%, which is less than in tap water.

$$Ca^{2+} + CO_3^{2-} \rightarrow CaCO_3 \tag{8}$$

$$HCO_3^- + HO^\bullet \rightarrow H_2O + CO_3^{-\bullet} \tag{9}$$

$$CO_3^{2-} + HO^\bullet \rightarrow HO^- + CO_3^{-\bullet} \tag{10}$$

$$Cl^- + HO^\bullet \rightarrow HO^- + Cl^{\bullet-} \tag{11}$$

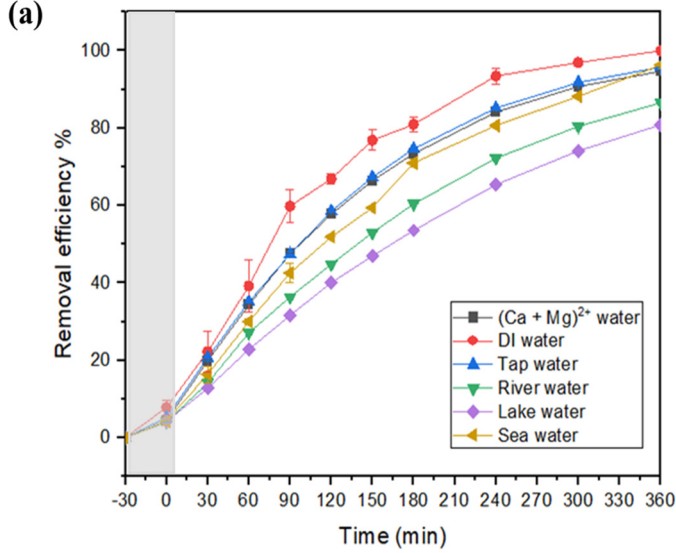

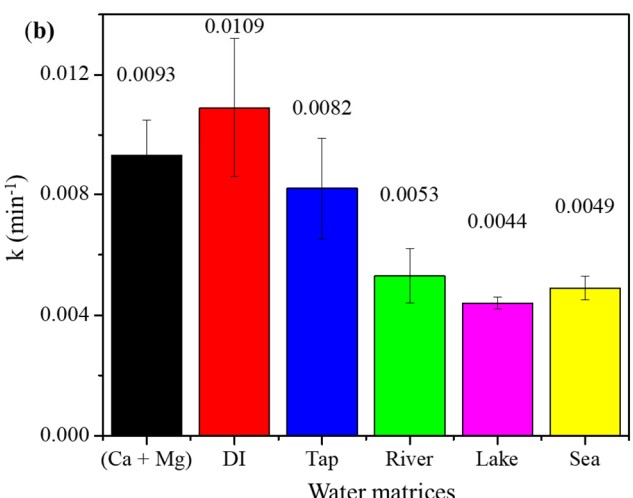

**Figure 4.** Degradation of CBZ in different waters: (**a**) Removal efficiency; (**b**) kinetic degradation. (DI = ultrapure water.)

### 3.2. Mineralization Studies

Ideally, the final by-products of degradation would be $CO_2$ and $H_2O$, and the TOC value would be close to zero. Therefore, TOC is one of the important parameters to monitor in the decomposition of organic compounds. The TOC remaining in the solution after degradation was measured (Figure 5). The mineralization efficiency was 31.7–71.3% after 6 h of irradiation under different conditions. Oxidative photodegradation involves two strong photogenerated oxidizing agents, $h^+$ and $HO^\bullet$ radicals, to evolve into carbon dioxide ($CO_2$) (Equation (12)) [35]. Although the reason is uncertain, it is possible that the presence of acetic acid during decomposition caused mineralization due to a process similar to its formation from malic acid [36] (Equations (13) and (14)).

Indeed, Figure 5 shows that the removal efficiencies had a similar trend to the percentage of photodegradation. In pH 2, this value was the highest (71.3%) while the lowest TOC value belonged to the lake condition (31.7%). As mentioned before, the high amount of DOM in lake or river water significantly decreased the CBZ decomposition, as well as TOC, compared to other types of water. In contrast, at pH 2, photogenerated $h^+$, free to operate without being prevented by any free radical, was also the main reason leading

to the highest mineralization value [25]. Moreover, with the presence of agents such as inorganic anions and HA, the TOC value had a significant downward trend. On the other hand, with the addition of the inorganic anion, $NO_3^-$, the TOC value tended to increase. When a concentration of $NO_3^-$ (50 mg $L^{-1}$) was added, the TOC value increased slightly from 55.79% to 66.2%. It was explained that, in addition to the free-running photogenerated $h^+$, $NO_3^-$ also generated more $HO^\bullet$ radicals, which were one of the beneficial factors for the decomposition process [12].

$$CH_3COOH + h^+ \rightarrow CO_2 \tag{12}$$

$$COOH\text{-}CH_2\text{-}COOH \rightarrow H_2C^\bullet\text{-}COOH \tag{13}$$

$$H_2C^\bullet\text{-}COOH \rightarrow CH_3COOH \tag{14}$$

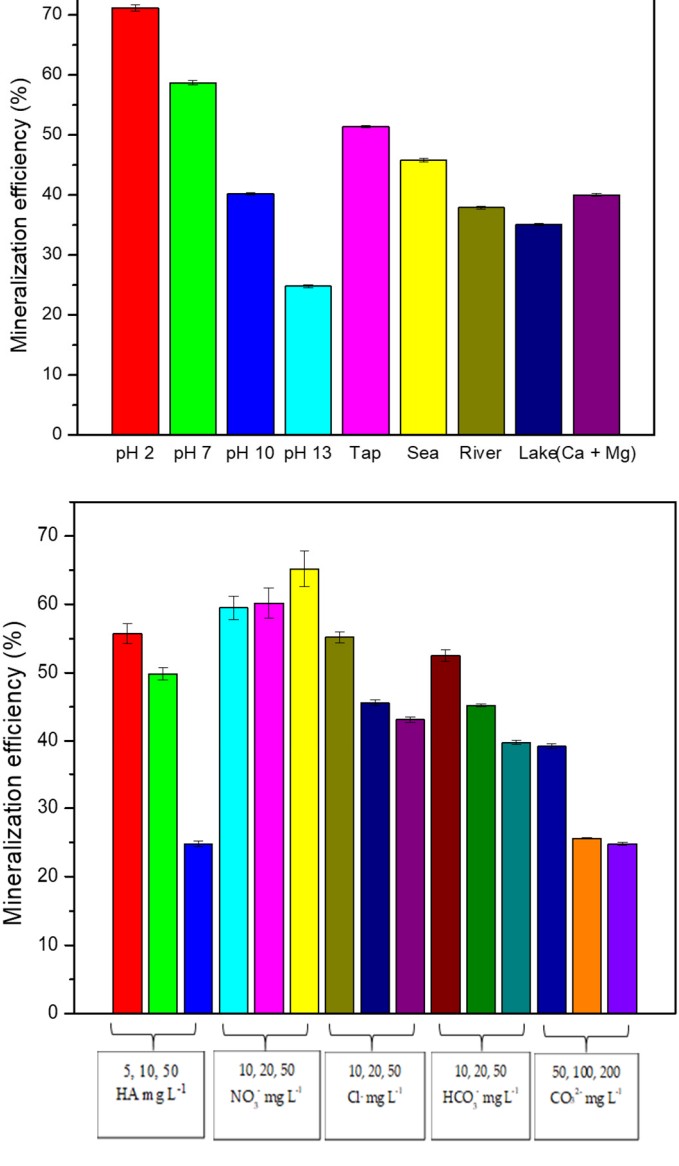

**Figure 5.** Mineralization efficiency of 5 mg $L^{-1}$ carbamazepine under different conditions.

## 3.3. Mechanism and Proposed Pathways

At the beginning of the experiment, only the peak of BBZ was observed before the light was turned on (Figure 6). Then, the peak area of CBZ decreased dramatically and new

lower peaks appeared, confirming an efficient degradation of CBZ and the formation of by-products. The retention time of CBZ was determined at 7.737 min, while the retention time of the by-products was detected at 1.157, 1.694, 6.568, 7.037, and 10.55 min. This was illustrated in the mechanism (Figure 6) and detected by MS (mass spectrometry).

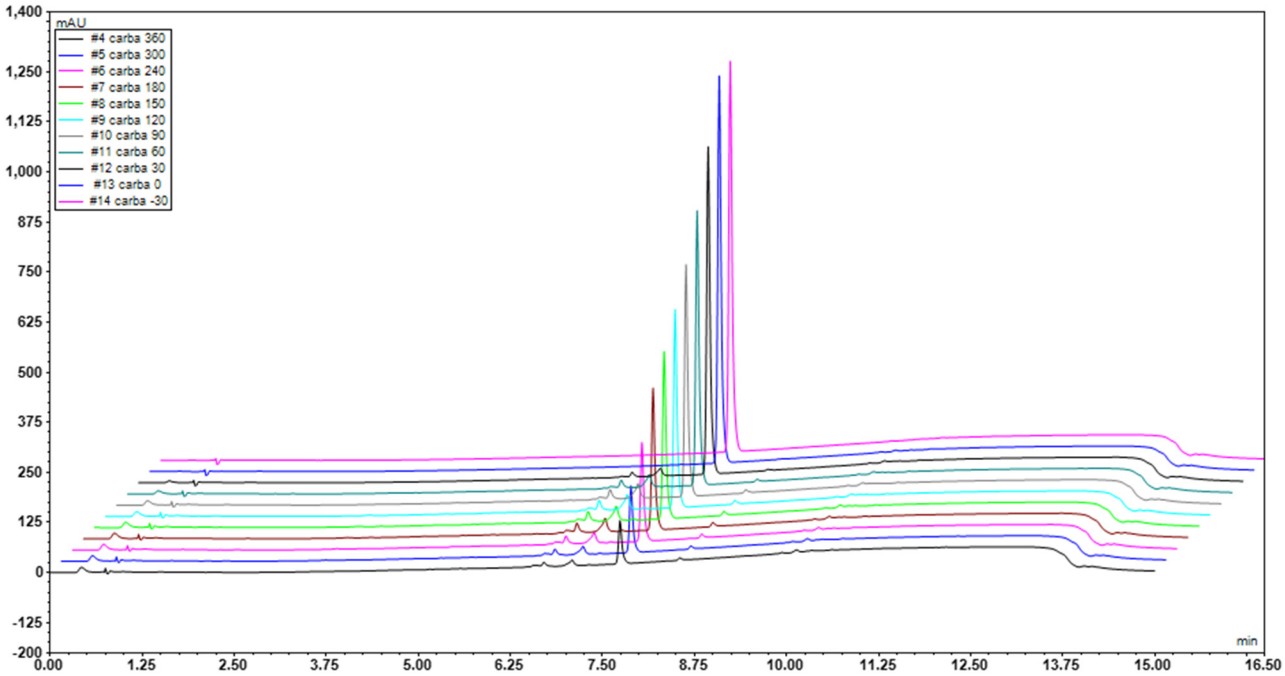

**Figure 6.** Chromatogram by HPLC of the degradation process of CBZ.

The mechanism of the by-product formation is shown in Figure 7 and is hereafter discussed in comparison to the literature. During the photodegradation process of CBZ, there are two main pathways. First, (Phenytoin) is formed by the hydration of CBZ ($m/z = 253 = 237 + 16$) [9]. The by-product at $m/z = 251$ could be the predominant secondary degradation by-product generated from 10, 11-epoxycarbamazepine due to the intramolecular reaction [12]. In addition, the by-product at $m/z = 251$ was oxidized to form a by-product at $m/z = 210$. After that, the epoxy bond of 10, 11-epoxy carbamazepine was attacked by two $h^+$ or $O_2^{\bullet-}$ radicals, which led to the appearance of 10-1(1aminovinyl)-10,10a-dihydroacridine-9-carbaldehyde, but there was also intramolecular transformation, so the molecular weight could not change [25]. Then, the by-product at $m/z = 208$ was produced as the ring contraction combined with the reduction (-$CONH_2$) of (1,1(1-aminovinyl)-10, 10a-dihydroacridine-9-carbaldehyde) and was oxidized again to form acridine (180) and acridone (196) [12]. Another means of by-product production is the formation of 10-hydroxy carbamazepine by the attack of $HO^{\bullet}$ to the olefin bond in the carbamazepine ring, which could be constructed into 10,11-dihydro-10,11-dihydroxycarbamazepine when this bond interacts with $HO^{\bullet}$ groups again. Finally, the above by-products undergo another ring-opening reaction and the product detected is acetic acid ($m/z = 60$), which can be mineralized to $CO_2$ and $H_2O$ via oxidation. Combined with the TOC removal efficiency (71.27%), this demonstrates that most of the by-products can be removed into small inorganic molecules by the $TiO_2$ photocatalyst. Some other by-products, such as acridont-hydroxide ($m/z = 226$), 3-hydroxybenzoic acid ($m/z = 137$), and by-products at $m/z = 115$, were found [12,37], but in this study, signals with these masses were not detected.

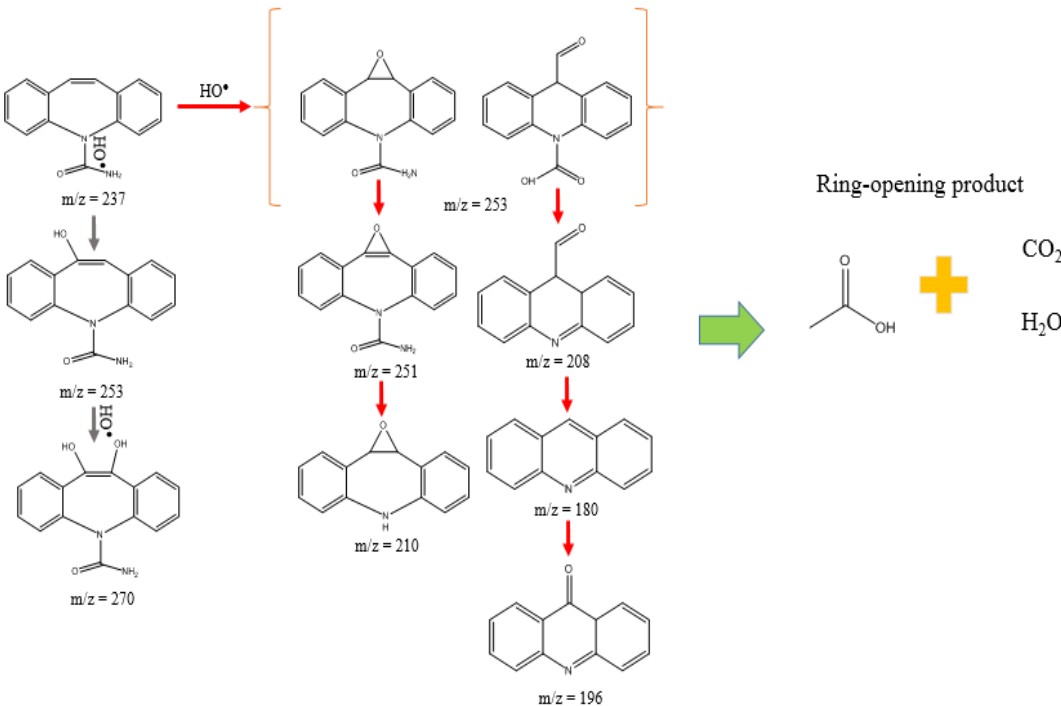

**Figure 7.** Proposed by-products of CBZ degradation in visible light.

As shown in Figure 8, the peak intensity of CBZ and other by-products was obtained at different masses. Specifically, in the initial condition (t = −30 min), the peak at $m/z = 237$ was the dominant peak with the highest signal. After 60 or 180 min of degradation, some by-products appeared as revealed by new mass peaks at $m/z$ equal to 60, 180, 196, and 270. The name, formula, and structure of the by-products are shown in Table 2.

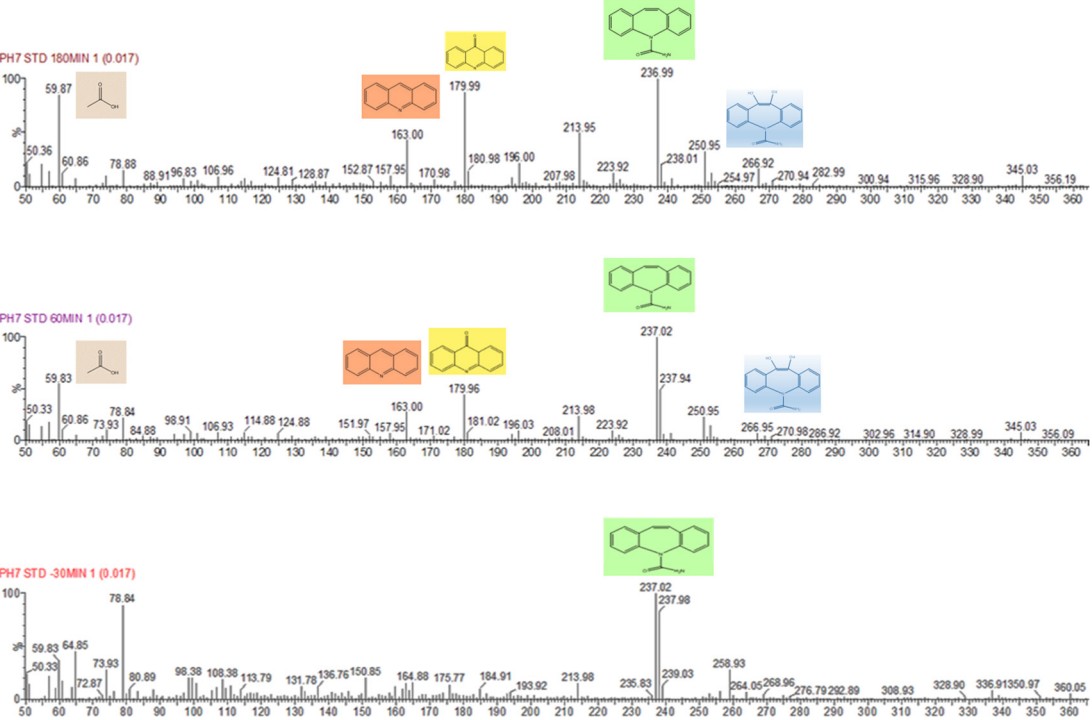

**Figure 8.** Peaks mass of carbamazepine and some by-products.

**Table 2.** Identification of by-products in the photodegradation of CBZ.

| Name | Formula | *m/z* | Structure |
|---|---|---|---|
| Acetic acid | $C_2H_4O_2$ | 60 | |
| Acridine | $C_{13}H_9N$ | 180 | |
| Acridone | $C_{13}H_9NO$ | 196 | |
| Acridine-9-carboxylic acid | $C_{14}H_9NO_2$ | 223 | |
| Phenytoin | $C_{15}H_{12}N_2O_2$ | 253 | |
| 10,11-dihydro-10,11-dihydroxycarbamazepine | $C_{15}H_{14}N_2O_3$ | 270 | |

### 3.4. Reusability of Photocatalyst

The resulting composites were designed mainly to allow their easy separation after the process [5]. After about 90 cycles, the degradation efficiency in the subsequent process experienced a slight decrease under the same condition as the initial one (ultrapure water, pH = 7). This value decreased from 100% after 360 min to 91.26% with a decrease in the mass of the $TiO_2$ photocatalyst of about 5% (from 41 g to 39 g). It is clear that $TiO_2$ in the ceramic form is extraordinarily stable and that the catalyst was occasionally chipped and flushed from the system. The photocatalytic activity can be regenerated by washing with HCl 1 M. The washing step was performed by cycling with HCl 1 M for 30 min to eliminate the impurity sticking on the surface of the material, and then rinsing again with 1 L of ultrapure water [31]. Even though this led to a small decrease, the photocatalyst was purified and its activity also stabilized (Figure 9). In addition, the morphology of the surface of PFS-01 before and after the experiment was observed by a scanning electron microscope (SEM), and there was no change in the morphology of the surface (Figure S1). The XRD

results (Figure S2) showed a mixture of anatase, brookite, and rutile forms in the material and the XRD spectrums are similar for the material before and after the experiment.

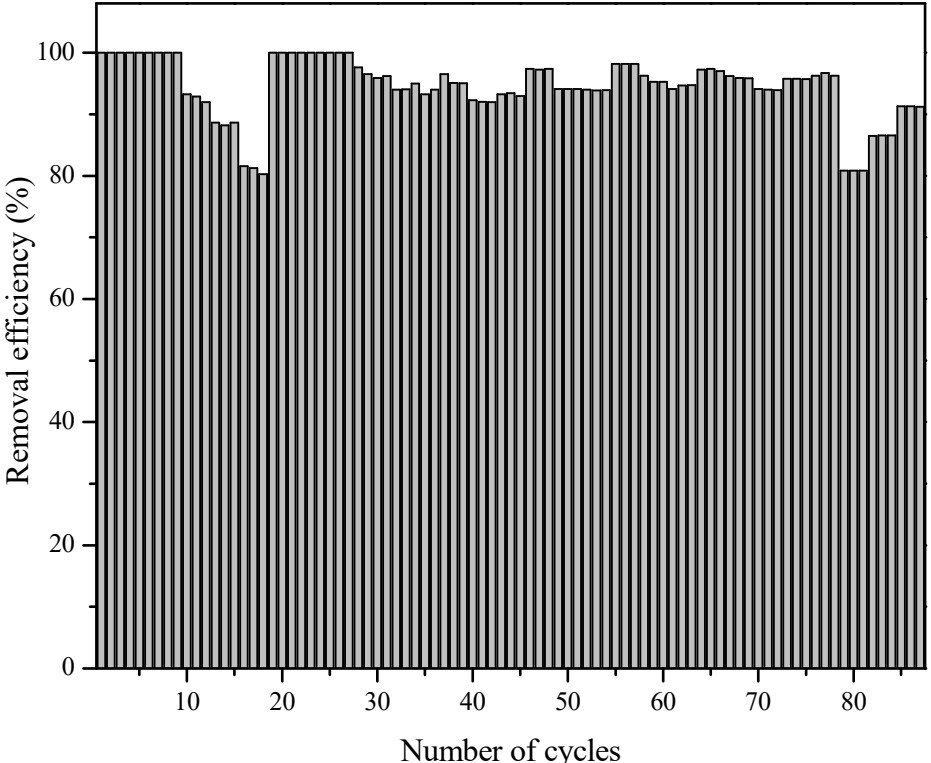

**Figure 9.** Reusability of the photocatalyst.

### 3.5. Prediction of By-Product Toxicity

Based on the analyzed results of Protox II, most of the by-products were found to be as toxic as or less toxic than carbamazepine, indicating the effectiveness of the proposed method in removing carbamazepine antibiotics from water. Only one by-product, acridine 9-carboxylic acid, was more toxic than CBZ (Table 3). These results are consistent with other studies that showed that the by-products of carbamazepine were Acridine; Acridone, which was found have an equal toxicity to carbamazepine; and/or Acridine-9-carboxylic acid, which was found to have a higher toxicity than CBZ [38–40]. Some by-products such as Acridone and Acridi are not only harmful to people but are also toxic to the organisms living in the water. Furthermore, the annual amount of carbamazepine released into the environment is extremely large, indicating that it is necessary to develop a method to remove it from the domestic environment.

**Table 3.** The toxicity of carbamazepine and by-products.

| Name | Predicted Toxicity Class |
|---|---|
| Carbamazepine | 4 |
| Acetic acid | 4 |
| Acridine | 1 |
| Acridone | 4 |
| Acridine-9-carboxylic acid | 5 |
| Phenytoin | 3 |
| 10,11-dihydro-10,11-dihydroxycarbamazepine | 4 |

### 4. Conclusions

This study shows that TiO$_2$ in a ceramic form (PSF-01 solid flake) is effective in removing CBZ under sunlight. The photocatalytic degradation of CBZ is enhanced under

acidic conditions with 100% decomposition after only 120 min of irradiation. The presence of $NO_3^-$ ion enhances the decomposition rate of CBZ, while the presence of other ions, including $Ca^{2+}$, $Mg^{2+}$, and $NH_4^+$, humic acid, and different water matrices decreases the degradation rate. In addition, the $TiO_2$ ceramic material exhibited good reusability and stability over a long period of use. Six main by-products were successfully identified by mass spectroscopy and their toxicities were identified by ProTox-II. Most by-products had similar toxicities to the parent compound but were generated at lower concentrations. Therefore, toxicity assessments should also be evaluated by practical tests to obtain accurate results instead of being predicted by the ProTox-II software. In the next step, the degradation of mixed compounds that usually exist in the real scenario will be studied before being applied in a larger pilot.

**Supplementary Materials:** The following supporting information can be downloaded at: https://www.mdpi.com/article/10.3390/w15081583/s1, Figure S1: SEM image of PFS-01 before and after experiments; Figure S2: XRD analysis of PFS-01 before and after experiments; Figure S3. UV Vis spectrum of PFS-01 material; Table S1. Reaction rate constant followed pseudo-first-order kinetics.

**Author Contributions:** T.H.N.: Conceptualization, experiment, formal analysis, writing—original draft preparation, and methodology. V.T.K.: Methodology and formal analysis. C.T.V., N.T.K.O., N.T.V.A. and L.H.L.: Funding acquisition, writing: review and editing, and visualization. N.N. and S.O.: Resources, supervision, validation, and writing—review and editing. B.V.H.: Conceptualization, methodology, supervision, validation, visualization, and writing—review and editing. All authors have read and agreed to the published version of the manuscript.

**Funding:** This research was funded by the University of Science and Technology of Hanoi (USTH), Vietnam Academy of Science and Technology (VAST) for Emerging Research Group (Decision number: 1077/QĐ-ĐHKHCN).

**Data Availability Statement:** Data are available from the corresponding author upon request.

**Acknowledgments:** This research was supported in part by the Kurita Asia Research Grant (Grant number: 22Pvn009-R3) provided by Kurita Water and Environment Foundation. This paper is also a contribution to the LOTUS International Joint Laboratory (http://lotus.usth.edu.vn, accessed on 16 April 2023) and the French National Research Institute for Sustainability Development (IRD). The authors also thank Nguyen Luong Lam and the AMSN department for the XRD measurement.

**Conflicts of Interest:** The authors declare no conflict of interest.

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
