# Peer review of "Removal of Carbamazepine in Aqueous Solution by TiO2 Ceramic Photo-Catalyst under Simulated Solar Light: Kinetics, Effects of Environmental Factors and Degradation Pathways"

_water, doi:10.3390/w15081583_

Round 1
Reviewer 1 Report
The author intends to degrade carbamazepine (CBZ) from an aqueous solution by using ceramic titanium dioxide instead of micro and nano structured titanium dioxide. I recommend this paper be published in Water. Nevertheless, I still have some comments or questions, it is suggested the authors could address these points before the final acceptance.
Comments:
1:Figure 6 in the manuscript cannot be displayed
2:The Figure 8 in the manuscript is too large to cover the text.
3:Why did the author use ceramic titanium dioxide instead of micro and nanostructured titanium dioxide to degrade carbamazepine? Please provide a specific explanation or comparison of the results.
Author Response
Dear Dr. Divyapriya Govindaraj, Dr. Sanjeeb Mohapatra
Dear Reviewers,
Thank you for giving us the opportunity to improve the quality of our manuscript “Removal of carbamazepine in aqueous solution by TiO2 ceramic photo-catalyst under simulated solar light: Kinetics, effects of environmental factors and degradation pathways” for publication in the special issue “Advances of Photocatalytic Application in Water and Wastewater Treatment” which belongs to the section of wastewater treatment and Reuse in the Water Journal. We highly appreciate the time and efforts that you and the reviewers dedicated to providing feedback on our manuscript. Your comments were valuable and high scientific impacts that help us to improve a lot for the revised version. Indeed, the authors have carefully considered all the comments and we have incorporated most of the suggestions made by the reviewers. Please see below, in red, for a point-by-point response to the reviewers’ comments. All modifications in the manuscript have been marked up using the “Track Change” function.
We hope the manuscript after careful revisions meet your high standards.
Once again, thank you for taking time out of your busy schedule to help us improve the quality of our manuscript. The authors welcome further constructive comments if any.
Sincerely yours,
BUI Van Hoi (PhD)
Department of Water – Environment – Oceanography
University of Science and Technology of Hanoi
A21 building, 18 Hoang Quoc Viet, Cau Giay, Hanoi
Tel:(+84) 2 437 917 206; Cell phone: (+84) 982 875 883
Website: http://usth.edu.vn

Reviewer 2 Report
In this manuscript, authors used monolithic titanium dioxide ceramic photocatalyst to degrade CBZ and study the degradation kinetics at different conditions under simulated solar lights.
The work is appropriately contextualized and the aim is clearly defined.
From my point of view, this manuscript is within the scope of Water.
Next, there are some comments that should be taken into account before considering this manuscript for publication.
Materials. Main characteristics of the TiO2 ceramic catalyst should be included in this section, not only the reference to other paper, as the use of this photocatalyst is one of the main novelties of the paper.
L146. include only the reference number, as in previous references.
Fig.2., 3 and 4 the shadow area in the graph seems to have been shift.
This figure 2 and all the others should include the bar indicating the standard deviation (the tests have been performed 3 times according to the experimental section).
L160. The kinetic rate for pH2 is not the same than in Figure 2
Fig 3. Please use the same colors in a and b graphs. Moreover, please, check the data. The evolution of removal efficiency in presence of 50mg/l of HCO3- and CO3-- seems to be very different to have these similar degradation rate. Also for the graphs in presence of 50mg/l of HA.
Fig.6 should be replaced. The information is not readable
Fig. 8. Some of the images in the Figure seems to be on the text, and make difficult the reading of the paper
In table 3, not all the by-products detected in table 2 were included. Please, complete the table and explain better the toxicity analysis.
The references should be revised carefully and the name of the full list of authors should be included.
Author Response

(The authors gave the same response as above.)

Reviewer 3 Report
This study reports the application of a small pilot of TiO2 in the ceramic form for removing carbamazepine (CBZ) from an aqueous solution under simulated sunlight. This research is meaningful. However, some modifications are required before publication, as follows:
1. The microstructures of TiO2 should be investigated via SEM.
2. The XRD pattern of TiO2 should be measured to confirm its phases.
3. The irradiation light energy intensity of simulated solar light should be measured and provided.
4. More significant photoresponsive catalysts should be introduced to keep abreast of the latest research trends. e.g.: Chem. Eng. J., 2023, 455, 140943; Adv. Fiber Mater., 2022, 4, 1620; Separation and Purification Technology, 2023, 304, 122401; Water 2022, 14(23), 3808.
5. The nanofiber-based catalysts show great advantages in the treatment of organic pollutants, e.g., Adv. Fiber Mater., 2022, 4, 1069; Adv. Fiber Mater., doi: 10.1007/s42765-022-00253-5; Adv. Fiber Mater., 2022, 4, 1278; Adv. Fiber Mater., 2022, 4, 573; These works are informative for readers.
6. Specific surface area is very important for performance of photocatalyst. The N2 adsorption/desorption isotherms and pore size distribution curves should be provided.
7. How about the stability of this material in this system?
Author Response

(The authors gave the same response as above.)

Reviewer 4 Report
In this manuscript, the results of this research are conveyed thoughtfully and completely, and they are consistent with the experimental findings. However, the authors failed to explain and draw out the novelty of the work, this aspect needs to be improved. This work is worthwhile to be publish in this journal after major revision. The following issues should be addressed:
1. Introduction is well-organized but the importance and novelty of the research should be highlighted and more clearly stated. The authors should give some examples of works in the bibliography, to clear the advantage of their work in comparison with those works.
2. Maybe the author should compare their results clearly with other reported works, highlighting the advantage and disadvantages of their novel composite.
3. The authors are responsible for the English, which should be polished throughout the manuscript to clear some minor typo/grammar errors.
4. Introduction part, if possible, some important and relative reports references could help:
https://doi.org/10.1007/s10904-022-02389-8
https://doi.org/10.1021/acsomega.1c03735
https://doi.org/10.1016/j.heliyon.2022.e09652
https://doi.org/10.1016/j.surfin.2022.102006
https://doi.org/10.3390/ma16062170
5. More information on the light source that the author used in the study and the filter used.
6. Experimental part. Please indication initial concentration of carbamazepine. And what is the average concentration of carbamazepine in wastewater that should be clean up?
7. Authors did not performed experiments on water purification using real wastewater. It is recommended to performed experiments on real wastewater, since there are many components that can significantly affect both catalytic properties and contaminate the catalyst.
8. Authors should discuss how prepared composites can be used in real experiments. Сan the composite contaminate water, and does it make it dangerous for human consumption. How can the composite be removed from the water after purification?
9. What is the optical band gap of the prepared material? Experimental evidences are needed?
10. authors predicted the degraded products, please explain how? An appropriate reaction mechanism of the degraded products should be discussed.
11. Stability tests are very important for any material performing as a photo-catalyst. Any morphological robustness, chemical compositional or oxidation state changes occur after photo-catalysis or not? Need experimental evidences in support of stability.
Hence, I recommend it accepted for publication after Major revisions.
Author Response

(The authors gave the same response as above.)

Round 2
Reviewer 1 Report
After the author's revision of the manuscript, now, I recommend this paper be published in Water thout any modifications.
Author Response
Thank you so much for your grateful advice, we acquired a huge knowledge from your comments.
Reviewer 3 Report
1. The current photocatalytic results should also be compared with the reported studies. 2. Error bars should be added in these test curves in Fig. 5. 3. More significant photoresponsive catalysts should be introduced to keep abreast of the latest research trends. e.g.: Chem. Eng. J., 2023, 455, 140943; Water 2022, 14(23), 3808, Adv. Fiber Mater., 2022, 4, 573; Adv. Fiber Mater., 2022, 4, 1620, 4. The active radial trapping experiments as well as ESR test must be added to confirm the decision roles of different active species 5. Pollutant degradation kinetics (first-order, and second order) studied.Author Response
Thank you a gain for your valuable comments. Please find in the enclosed find our responses
Best regards,

Reviewer 4 Report
Accepted in the present form
Author Response

(The authors gave the same response as above.)
